# Lower Survival and Increased Circulating Suppressor Cells in Patients with Relapsed/Refractory Diffuse Large B-Cell Lymphoma with Deficit of Vitamin D Levels Using R-GDP Plus Lenalidomide (R2-GDP): Results from the R2-GDP-GOTEL Trial

**DOI:** 10.3390/cancers13184622

**Published:** 2021-09-15

**Authors:** Carlos Jiménez-Cortegana, Pilar M. Sánchez-Martínez, Natalia Palazón-Carrión, Esteban Nogales-Fernández, Fernando Henao-Carrasco, Alejandro Martín García-Sancho, Antonio Rueda, Mariano Provencio, Luis de la Cruz-Merino, Víctor Sánchez-Margalet

**Affiliations:** 1Clinical Laboratory, Department of Medical Biochemistry and Molecular Biology, School of Medicine, Virgen Macarena University Hospital, University of Seville, 3139 Seville, Spain; cjcortegana@us.es (C.J.-C.); pilarm.sanchez.sspa@juntadeandalucia.es (P.M.S.-M.); fhcarrasco@us.es (F.H.-C.); 2Oncology Service, Department of Medicines, School of Medicine, Virgen Macarena University Hospital, University of Seville, 3139 Seville, Spain; npalazoncarrion@gmail.com (N.P.-C.); esteban.nogales@gmail.com (E.N.-F.); 3Department of Hematology, Hospital Universitario de Salamanca, Instituto de Investigación Biomédica de Salamanca (IBSAL), Centro de Investigación Biomédica en Red de Cáncer (CIBERONC), 37001 Salamanca, Spain; amartingar@usal.es; 4Department of Hematology or Clinical Oncology, Costa del Sol Hospital, Málaga, 29600 Marbella, Spain; rueda.dominguez@gmail.com; 5Department of Clinical Oncology, Puerta de Hierro University Hospital, Majadahonda, 28001 Madrid, Spain; mprovenciop@gmail.com

**Keywords:** diffuse large B-cell lymphoma, immune system, MDSC, Treg, PD-1, vitamin D, immunosuppression, lenalidomide

## Abstract

**Simple Summary:**

Diffuse large B-cell lymphoma (DLBCL) is an aggressive, heterogeneous non-Hodgkin B lymphoma, that 35% of cases produces relapsed/refractory (R/R) disease. At this point, the search of prognostic and predictive factors in DLBCL is indispensable through key biomarkers. Recently, we have found that the level of circulating MDSCs is a good marker of survival in a translational study based on the R2-GDP-GOTEL study (EudraCT Number: 2014-001620-29). Since Vitamin D is a known regulator of the immune system, we aimed to assess the evolution of circulating suppressor cells MDSCs, Tregs, and inhibited T cells in patients with deficit of vitamin D (<15 ng/mL) and sufficient levels of vitamin D (>15 ng/mL). We observed a reduction in blood suppressor cells in the group with normal vitamin D levels, but not in patients with vitamin D deficit, supporting that R2-GDP and vitamin D may have immunomodulatory functions that favors a better clinical evolution.

**Abstract:**

The search of prognostic factors is a priority in diffuse large B-cell lymphoma (DLBCL) due to its aggressiveness. We have recently found that the level of circulating MDSCs is a good marker of survival in a translational study based on a trial (EudraCT Number: 2014-001620-29), using lenalidomide combined with R-GDP (rituximab plus gemcitabine, cisplatin, and dexamethasone). Since Vitamin D is a known immunomodulator, we have studied blood levels of these cell populations comparing patients with deficit of vitamin D levels (<15 ng/mL with those with normal levels >15 ng/mL. Mann–Whitney U test was used to compare cells distributions between groups, Wilcoxon test to compare cells distribution at different times and Spearman test to measure the association between cell populations. Patients with vitamin D deficit maintained the increased level of immune suppressor cells, whereas we observed a depletion of all immune suppressor cells in patients with normal vitamin D levels. In conclusion, we have confirmed the importance of vitamin D in the response to treatment in R/R DLBCL, suggesting that vitamin D deficit may be involved in the immune deficit of these patients, and thus, vitamin D supplementation in these patients may help to obtain a better response, warranting further investigation.

## 1. Introduction

Diffuse large B-cell lymphoma (DLBCL) is the most common aggressive non-Hodgkin lymphoma (NHL) subtype, representing between 30% and 40% of cases every year [1]. One-third of DLBCL patients develops relapsed/refractory (R/R) disease after standard rituximab-based therapies. The second-line treatment consists of salvage chemotherapy followed by autologous stem cell transplantation (CT+ASCT), but some patients finally relapse or are not eligible for transplantation due to ineffective salvage treatment. Thus, a number of alternatives to CT+ASCT have been proposed [2].

Due to the aggressive behavior of the disease and its different clinical outcomes, some prognostic factors have been evaluated, including (but not limited to) vitamin D and the immune system [3]. Vitamin D is a liposoluble vitamin that exists in two major forms: ergocalciferol (vitamin D2) and cholecalciferol (vitamin D3). Ergocalciferol is mainly synthetized by plants from ergosterol and can only be obtained by food intake. However, while a small proportion of cholecalciferol is taken from dietary, the rest of this vitamin form (around 80% approximately) is produced endogenously from 7-dehydrocholesterol in the skin by the action of ultraviolet (UV) light [4]. Both ergocalciferol and cholecalciferol are biologically inactive forms and are first converted to calcidiol (25-hydroxyvitamin D) in the liver and later to calcitriol (1,25-dihydroxyvitamin D) in the kidney [4,5].

The role of vitamin D in some biological processes is well known, like the bone metabolism, or the calcium homeostasis and regulation [5]. However, its role in cancer is controversial, as serum vitamin D or vitamin D supplementation could play a key role in the prevention and treatment of many malignancies [6,7,8,9], although those promising findings were not confirmed in other studies [10,11]. Specifically, vitamin D deficiency has been extensively reported to be a negative prognostic factor in B-cell lymphomas (including DLBCL) [12,13,14,15], in which rituximab and lenalidomide are commonly used [16,17]. In vitro data suggest that supplementation with vitamin D may enhance rituximab and lenalidomide-mediated cytotoxicity in these hematologic malignancies [18,19], causing an improvement in the cytotoxicity of immune cells, such as NK lymphocytes [20] or macrophages [21]. In R/R DLBCL, rituximab combined with lenalidomide (R2) has been used alone [22] or combined with GDP (gemcitabine, cisplatin, and dexamethasone) [23], obtaining promising responses. In the same line, R2-GDP was used as an alternative schedule in R/R DLBCL patients from the study “Phase II clinical trial to evaluate the combination of lenalidomide with R-GDP in patients diagnosed with R/R DLBCL not candidates for high-dose chemotherapy and hematopoietic progenitor cell transplantation” (EudraCT Number: 2014-001620-29), performed by the Spanish Lymphoma Oncology Group (GOTEL), reporting promising objective response rates (ORR) [24,25] and the depletion of immune suppressor cells in patients with an overall survival (OS) longer than 24 months [26].

In recent decades, a great interest has emerged in cancer concerning to the potential role of the immune system and its modulation by vitamin D, since vitamin D receptor (VDR), a member of the nuclear receptor superfamily, is expressed to a greater or lesser extent on both innate and adaptive immune cells, such as neutrophils, dendritic cells (DCs), or T and B lymphocytes [27]. Vitamin D exerts its modulatory action over the immune system by binding to the VDR. On the one hand, although VDR expression decreases during the monocyte differentiation into macrophages or DCs, 1,25(OH)_2_D_3_ enhances innate immune responses by increasing the production of antimicrobial proteins and both chemotactic and phagocytotic responses of macrophages [28], and by reducing the levels of CD34+ myeloid cells, that are precursors of myeloid-derived suppressor cells (MDSCs) [29], found in high concentrations in both the tumor microenvironment (TME) and peripheral blood (PB) [30].

By contrast, VDR is highly expressed during the T cell activation and indirectly stimulates the polarization of T cells from Th1 and Th17 to Th2 phenotype via antigen-presenting cells (APCs) by inhibiting the expression of MHC-II or co-stimulatory molecules, and directly by inhibiting the production of IL-12, IFN-γ, or IL-17 and stimulating IL-4, again leading the polarization from Th1 and Th17 to Th2 cells, as well as allowing the development of regulatory T cells (Tregs) by increasing FOXP3 and IL-10 production [28]. In cancer, suppression of immune responses can be carried out not only by MDSCs or Tregs, but also by both CTLA-4 and PD-1 pathways [31]. 1,25(OH)2D3 is able to enhance CTLA-4 expression in T cells [32]. However, the modulation of PD-1 remains unclear since vitamin D has been demonstrated to increase the PD-L1 expression (and, therefore, PD-1/PD-L1 interaction) and decrease the expression of co-stimulatory receptors, such as CD80 and CD86 in healthy individuals [33] and enhance PD-1 expression in Crohn’s disease [34], while this inhibitory expression has been reduced in both helper and cytotoxic T cells with vitamin D treatment in cystic fibrosis [35].

In this work, our purpose was to evaluate the evolution of different immune suppressor cells (MDSCs, Tregs, and both exhausted PD-1+OX40- and CTLA-4+OX40- T cells) throughout the R2-GDP treatment in R/R DLBCL patients with normal vitamin D (NVD) levels and vitamin D deficiency (VDD) (>15 ng/mL and <15 ng/mL, respectively), recruited in the R2-GDP-GOTEL study. In addition, we aimed to compare basal vitamin D levels of patients with those in healthy donors (HD) and to correlate initial vitamin D levels of patients according to OS at 24 months.

## 2. Results

### 2.1. Clinical Characteristics of R/R DLBCL Patients

From April 2015 to September 2018, the R2-GDP-GOTEL trial recruited 79 patients diagnosed of R/R DLBCL not candidates for ASCT from 18 Spanish hospitals. Major inclusion criteria included patients with R-CHOP-like treatments previously received and ECOG ≤ 1. They were treated with the R2-GDP schedule, based on lenalidomide, rituximab, gemcitabine, dexamethasone and cisplatin, and granulocyte colony-stimulating factor. 

Vitamin D levels could only be analyzed in 63 R/R DLBCL patients before treatment and in all HD (20). Data analyses in cycle 3 and EOI were lower due to adverse effects or death in some patients throughout the course of the treatment. A summary of the number of patients included in the data analysis is shown in Figure 1. 

### 2.2. Low Vitamin D Levels before R2-GDP May Predict a Poorer OS in R/R DLBCL

Serum vitamin D levels were significantly lower in R/R DLBCL patients before starting treatment compared with the age-matched HD (*p* < 0.001), as shown in Figure 2A. In addition, basal vitamin D was found higher in patients with an OS longer than 24 months after treatment (*p* = 0.010) compared with OS < 24 month patients, as shown in Figure 2B. 

### 2.3. Circulating MDSCs Decreased in R/R DLBCL Patients with Better Vitamin D Levels after R2-GDP

Both M-MDSC and G-MDSC populations decreased in blood after treatment in the NVD group comparing with basal (Figure 3A,B, respectively), with *p* = 0.047 (for M-MDSCs) and *p* = 0.020 (for G-MDSCs). Consequently, total MDSC levels were also remarkably reduced in the same group of patients (*p* = 0.002), as shown in Figure 3C. In the VDD group, M-MDSCs were increased at the end of the treatment (Figure 3A), but both G-MDSCs and total MDSCs remained constant (Figure 3B,C, respectively). Comparing both vitamin D groups, statistical differences were found in M-MDSCs and total MDSCs in the EOI determination (*p* < 0.001 in both cases).

In addition, significant negative correlations were found between basal vitamin D levels and G-MDSCs before treatment (rS = −0.315; *p* = 0.015), and between basal vitamin D and M-MDSCs after treatment (rS = −0.576; *p* = 0.016), as shown in Appendix A.

### 2.4. Peripheral Blood Treg Levels Were Significantly Reduced with R2-GDP in NVD Patients

Circulating Tregs were decreased in both vitamin D groups at the end of the induction with R2-GDP (Figure 4). However, this T cell subset was significantly reduced only in NVD patients (*p* = 0.008). Statistical differences were also found in cycle 3 (*p* = 0.048) and EOI (*p* = 0.004) between NVD and VDD patients.

In addition, negative correlations were found between basal vitamin D levels and Tregs before (rS = −0.274; *p* = 0.037) and after treatment (rS = −0.521; *p* = 0.013), as shown in Appendix A.

### 2.5. Inhibited PD-1+OX40− T Cells Were Notably Reduced after Treatment in Patients with Better Vitamin D Concentration

CD4+, CD8+, and total PD-1+OX40− T cell levels were significantly reduced after the R2-GDP schedule in NVD patients compared with basal determinations (*p* = 0.004 in CD4+, and *p* = 0.002 in both CD8+ and total PD-1+OX40− T cells), as shown in Figure 5A–C, respectively. By contrast, those T cell subsets were slightly increased in the VDD group. In addition, significant differences existed after treatment between NVD and VDD patients in CD4+ (*p* = 0.037), CD8+ (*p* = 0.047) and total PD-1+OX40− T cells (*p* = 0.013).

Negative correlations were found between initial vitamin D and EOI inhibited PD-1 T cells (Appendix A), but statistically significant only in the CD4+ subset (rS = −0.454; *p* = 0.030).

### 2.6. Exhausted CTLA-4+OX40- T Cells Were Significantly Depleted in NVD Patients Using R2-GDP Schedule

In line with PD-1+OX40- T cells, all inhibited CTLA-4+OX40− T cell levels were also reduced in blood after treatment comparing with basal (*p* = 0.027 for CD4+, *p* = 0.014 for CD8+, and *p* = 0.002 for total CTLA-4+OX40− T cells), as shown in Figure 6A–C, respectively. By contrast, CTLA-4 subset was significantly increased after R2-GDP therapy in VDD patients in both CD4+ (*p* = 0.003) and total CTLA-4+OX40− T cells (*p* = 0.021). Statistical differences between NVD and VDD groups were also identified at EOI in CD8+ (*p* = 0.015) and in both CD4+ and total CTLA-4 T cells (*p* < 0.001).

Again, significant negative correlations were found between basal vitamin D levels and EOI CTLA-4 T cells, as show in Appendix A: rS = −0.708 in CD4+ (*p* < 0.001), rS = −0.431 in CD8+ (*p* = 0.036), and rS = −0.735 in total exhausted CTLA-4+OX40− T cells (*p* < 0.001).

## 3. Discussion

In recent years, there has been a great enthusiasm due to the promising role of vitamin D in the prevention and immunomodulation of cancer. The first observation of the inverse relationship between the exposure to sunlight and cancer mortality was made in 1941 by Apperly et al. [36]. Since then, a lot of studies has been carried out supporting the potential role of vitamin D in the prevention and treatment of many types of cancer [37,38] as vitamin 1,25(OH)_2_D has been demonstrated to have anticarcinogenic properties [39]. Because of that, the use of calcitriol and vitamin D analogues has also been studied [40] but its clinical applications have been limited by hypercalcemia, the main toxic effect of vitamin D in blood [41,42]. 

In DLBCL, low concentrations of vitamin D have been found in patients at diagnosis and a deficiency of this vitamin type has been correlated with poorer clinical outcomes [14,18,43,44]. That is why we wanted to check the vitamin D levels in the patients with R/R DLBCL included in an intervention trial (the R2-GDP-GOTEL trial), using lenalidomide as immunomodulator. Our results were oriented in the same direction and we observed significant low levels of serum vitamin D in R/R DLBCL patients compared with age-matched healthy subjects, as well as a better OS after 24 months in patients with vitamin D levels higher than 15 ng/mL before treatment, which means that vitamin D deficiency could play a key role in the immunosuppression of the disease but, once the R2-GDP schedule began, could contribute to a worse long-term clinical outcome, considering the advanced age of the patients from the R2-GDP-GOTEL trial [26]. The lower vitamin D levels in patients compared with controls should be expected since these patients may not have as many open-air activities as healthy controls. It has been shown that vitamin D has beneficial effects by inducing programmed cell death and apoptosis, as well as stimulating cell differentiation. The elevated immunosuppression found in R/R DLBCL patients, and their high age implied the loss of control of normal processes, including a reduction in vitamin D, which could promote other different processes, such as the alteration of the immune balance, neo-angiogenesis, tumor escape, and the remodeling of the extracellular matrix.

Vitamin D has essential interactions with both innate and adaptive cells for normal immune functions, and impaired concentrations could produce dysregulated immune responses [45], thus confirming the immunomodulator role of vitamin D not only in healthy subjects but also in cancer [46]. In this sense, vitamin D was demonstrated to improve the cytotoxicity of rituximab and lenalidomide, and of some cell populations, as mentioned before [19,20,21,22]. Nevertheless, it is also important to consider the role of immune response suppressor cells or immune checkpoint pathways as targets to overcome the tumor-induced immunosuppression, such as MDSCs [47], Tregs [48], or both PD-1 and CTLA-4 pathways [49,50]. In this regard, our group previously reported the depletion of MDSCs, Tregs and inhibitory PD-1+OX40− T cells, as well as the increase in activated OX40+PD-1− T cells in R/R DLBCL patients with OS > 24 months from the R2-GDP-GOTEL trial [26].

Here, we show the results from the same trial but focused on the concentration of serum vitamin D before starting R2-GDP schedule and its impact on immune suppressor cells in patients with deficit of vitamin D (<15 ng/mL) or normal levels of vitamin D (>15 ng/mL). We found increased baseline levels of circulating M-MDSCs, G-MDSCs, and total MDSCs independently of the vitamin D concentration. However, when we monitored MDSCs during the course of the treatment we observed a remarkably depletion of these cell populations in the group of patients that had vitamin D levels >15 ng/mL, while MDSC populations were broadly unchanged in the group with deficit of vitamin D. Lenalidomide has inhibited MDSC-mediated immunosuppression in multiple myeloma [51] and vitamin D was demonstrated to promote the differentiation of MDSCs into mature cells without suppressive activities [52], as this vitamin reduces the levels of CD34+ immature myeloid cells, which are precursors of MDSCs [29].

The role of vitamin D over Tregs also seems to be clear since this substance has been demonstrated to promote the differentiation of Tregs in many diseases [53,54,55,56] and to inhibit the proliferation (but not their suppressive role) in IHV [57]. Today, it is accepted that the main role of Tregs in cancer is immunosuppressive [58] and, consequently, the interaction between Tregs and vitamin D has been little studied in this field. However, Karkeni et al., 2019 found that vitamin D did not affect the presence of regulatory T cells in breast cancer-bearing mice [59]. In our case, nor did low levels of vitamin D in patients affect to circulating Tregs before treatment, but the R2-GDP immunotherapy contributed to improve the immunosuppression in those with better levels of vitamin D, while Tregs were slightly depleted in the VDD group. To our knowledge, this is the first time that blood Tregs are reduced in cancer patients with better vitamin D levels, which could be explained by the use of the R2-GDP schedule.

Apart from the regulatory subset of T cells, we also studied other T cell subpopulations, such as PD-1 and CTLA-4 T cells, mainly involved in different pathways as immune checkpoint in cancer [49,50]. We obtained an important depletion of both PD-1+OX40- and CTLA-4+OX40− T cells in patients with better initial vitamin D levels using R2-GDP, while inhibited PD-1 T cells remained constant and CTLA-4 T cells were significantly increased in patients with worse vitamin D levels. Although vitamin D is known to module immune responses by up-regulating the expression of PD-1 in Crohn’s disease [34] and both PD-L1 and PD-L2 in cell-based models [60], vitamin D supplementation has also reduced the expression of PD-1 in CD4+ and CD8+ T cells in cystic fibrosis [35]. This last result is in the same line as our results, as we observed a depletion of PD-1+OX40− in both CD4+ and CD8+ T cell subset only in patients with higher serum vitamin D levels when R2-GDP therapy was used. By contrast, Jeffery et al., 2009, 2012, 2015 have extensively study the role of vitamin D over CTLA-4 and they concluded that vitamin D promoted the induction of Tregs expressing CTLA-4 surface marker [32,61,62]. However, in our case, the group of patients with better initial vitamin D concentrations achieved a significant reduction in the levels of inhibited CTLA-4+OX40− T cells through the use of the R2-GDP treatment.

## 4. Materials and Methods

### 4.1. Patients

We studied circulating immune suppressor cells and vitamin D levels of patients from the R2-GDP-GOTEL study (EudraCT Number: 2014-001620-29), a phase II clinical trial that recruited a total of 79 patients diagnosed of R/R DLBCL who were not high-dose CT and ASCT candidates from 18 Spanish hospitals between April 2015 and September 2018. They received the R2-GDP schedule, based on lenalidomide + R-GDP (rituximab, gemcitabine, dexamethasone, and cisplatin). An age-matched group of 10 women and 10 men was also recruited as healthy donors in July 2019 from the Virgen Macarena University Hospital (Seville, Spain). Median age were 71.3 years old in R/R DLBCL patients and 68.2 years old in the healthy group.

### 4.2. Flow Cytometry Analysis in Whole Blood Samples

Three peripheral blood analysis were carried out during the R2-GDP-GOTEL study: basal, cycle 3 and end of induction (EOI). MDSCs, Tregs, and inhibited PD-1+OX40− and CTLA-4+OX40− T cells were measured by flow cytometry using the FACSCanto II flow cytometry system (Becton Dickinson, Madrid, Spain) from EDTA-K3 tubes. Gates of cell populations are shown in Appendix A. M-MDSCs were gated as CD45^+^CD11b^+^CD33^+^HLA-DR^low/−^CD14^+^CD15^−^, G-MDSCs as CD45^+^CD11b^+^CD33^+^HLA-DR^low/−^CD14^−^CD15^+^, Tregs as CD4^+^CD25^high^CD127^low/−^, inhibited PD-1+ T cells as CD3^+^CD4^+^PD-1^+^OX40^−^ and CD3^+^CD8^+^PD-1^+^OX40^−^, and inhibited CTLA+ T cells as CD3^+^CD4^+^CTLA-4^+^OX40^−^ and CD3^+^CD8^+^CTLA-4^+^OX40^−^. Absolute cell number was calculated by multiplying flow cytometry percentages with total leucocyte count from hematologic count (Sysmex CS-1000, Barcelona, Spain). Total MDSCs were calculated as the sum of M-MDSC and G-MDSC counts, total PD-1 T cells as the sum of CD3^+^CD4^+^PD-1^+^OX40^−^ and CD3^+^CD8^+^PD-1^+^OX40^−^ T cell counts, and total CTLA-4 T cells as the sum of CD3^+^CD4^+^CTLA-4^+^OX40^−^ and CD3^+^CD8^+^CTLA-4^+^OX40^−^ T cell counts.

### 4.3. Monoclonal Antibodies

Antibodies were obtained from Becton Dickinson Immunocytometry Systems (BDIS, San Jose, CA, USA) and were used at the manufacturer’s recommended concentrations.

MDSCs: PerCP-Cy5.5 Mouse Anti-Human CD 45 (ref no. 564105), APC-Cy7 Rat Anti-CD11b (ref no. 557657), PE Mouse Anti-Human CD 33 (ref no. 555450), PE-Cy7 Mouse Anti-Human HLA-DR (ref no. 560651), FITC Mouse Anti-Human CD 14 (ref no. 555397), and APC Mouse Anti-Human CD 15 (ref no. 551376).

Tregs: Human regulatory T cell cocktail (ref no. 560249), including PerCP Mouse Anti-Human CD4, PE Mouse Anti-Human CD127, and FITC Anti-Human CD25.

PD-1 and CTLA-4 T cells: APC-Cy7 Mouse Anti-Human CD3 (ref no. 560176), PE-Cy7 Mouse Anti-Human CD4 (ref no. 557852), PerCP-Cy5.5 Mouse Anti-Human CD8 (ref no. 565310), APC Mouse Anti-Human PD-1 (CD279) (ref no. 558694), and PE Mouse Anti-Human CTLA-4 (CD152) (ref no. 557301).

### 4.4. Vitamin D Analysis in Serum

Basal vitamin D samples were collected in cryotubes from serum-separating tubes and stored at −80 °C. At the end of the R2-GDP-GOTEL study, all samples were measured in the same day by an automated chemiluminescence immunometric analysis using the Liaison^®^ (DiaSorin, Madrid, Spain) system.

### 4.5. Data Analysis

Statistical analysis and graphs were performed by GraphPad Prism 8.0.2 (GraphPad Software, San Diego, CA, USA). Normal distribution of analyzed variables was checked by watching histogram, box plot, Q-Q plot, and the outcomes of normality tests of Shapiro–Wilk and Kolmogorov–Smirnov.

Non-parametric tests were used due to absence of normality. Mann–Whitney U test was used to compare basal cell distributions of all patients vs. HD, and NVD vs. VDD patients in every determination (basal, cycle 3, and EOI). The Wilcoxon test was used to compare cell distributions of patients in basal vs. EOI determinations. Data shown are median and 95% confidence intervals. Bivariate correlations among cell populations were carried out using Spearman coefficient. *p* values ≤ 0.05 were considered for statistically significant differences.

## 5. Conclusions

In last years, efforts in cancer research have been mainly focused on the targeting of immune suppressor cells to overcome tumor resistance. As we know, some emerging prognostic factors in DLBCL are the immune system and vitamin D. This is the reason why we decided to evaluate the evolution of immune response suppressor cells, such as MDSCs, Tregs, and both inhibited PD-1+OX40− and CTLA-4+OX40− T cells in R/R DLBCL patients with deficit and normal levels of vitamin D levels using R2-GDP as an immunomodulator in the R2-GDP-GOTEL study (EudraCT Number: 2014-001620-29). In B-cell lymphomas, blood vitamin D levels were shown to affect both the course and progression of the disease by different mechanism of action, such as a possible systemic effects, effects on pharmacokinetics or the activation of the NF-κB pathway [63], and the use of vitamin D treatments increase the expression of some markers, such as CD25, FOXP3, CTLA-4, or PD-1. However, our results have shown increased levels of these immune cells in patients with deficit of Vitamin D, whereas patients with normal Vitamin D levels had a consistent decrease in these cells upon the R2-GDP treatment. On the other hand, the use of vitamin D supplementation could be promising since it may restore suppressor cells to normal values and improve outcomes in DLBCL. However, there is some controversy related to the cut-off dosage values and the correct substitution regimen, since they may result in renal failure and cardiac arrest due to hypercalcemia. In addition, some variables are implied, such as skin color, latitude, or season) [64]. In our case, basal vitamin D has been found to differentiate immature immunosuppressive cells. Nevertheless, because of the limited number of patients, further studies are required to confirm these results, even though the results are clear regarding patients with deficit of vitamin D. Despite this, we believe that the use of R2-GDP plus vitamin D supplements in R/R DLBCL patients may have a high potential which would need to be evaluated in future studies, as this combination could improve clinical results, also reflected not only in the boost of lenalidomide, rituximab, or immune cell cytotoxicities against tumor cells, but also in the depletion of immune suppressor cells in B-cell lymphomas, especially MDSCs and Tregs, which are currently considered as the main suppressors of the T cell-mediated immune response.

## Figures and Tables

**Figure 1 cancers-13-04622-f001:**
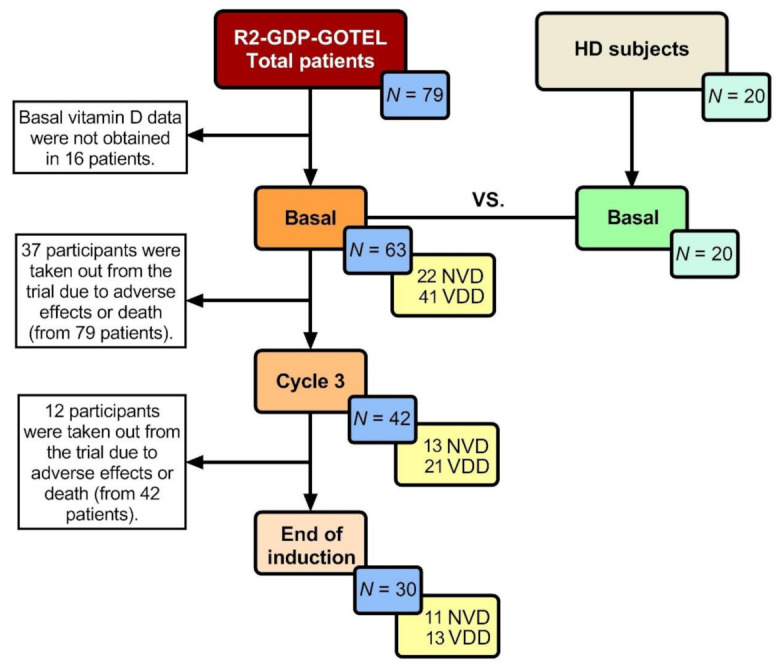
Number of patients analysed in every determination (yellow box). HD: Healthy donors; NVD: Normal vitamin D levels; VDD: Vitamin D deficiency.

**Figure 2 cancers-13-04622-f002:**
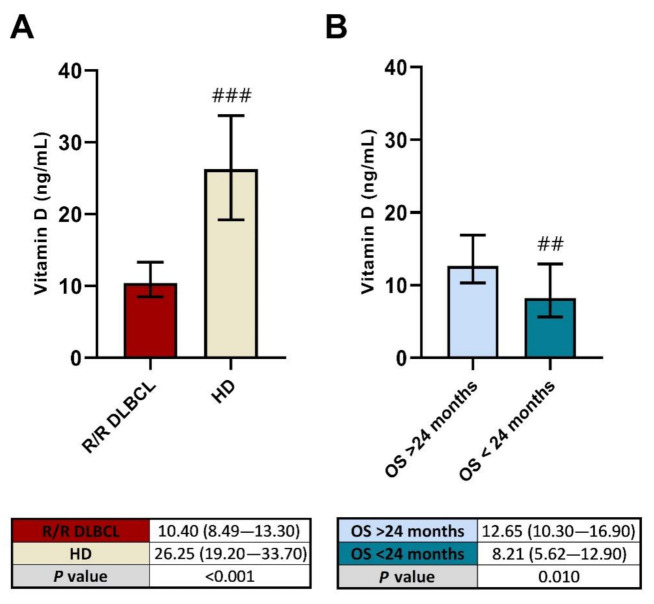
Basal vitamin D levels of R2-GDP-GOTEL patients compared with healthy donors (**A**) and according to the overall survival at 24 months (**B**). HD: Healthy donors; OS: Overall survival. ## and ###, statistically significant differences compared with the other group (*p* ≤ 0.01 and *p* ≤ 0.001, respectively).

**Figure 3 cancers-13-04622-f003:**
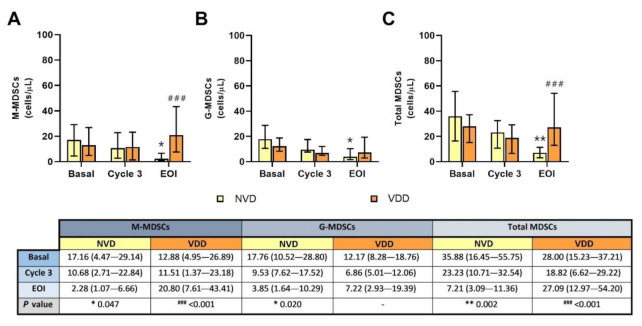
Circulating MDSC concentration according to normal vitamin D (NVD) levels and vitamin D deficiency (VDD) levels in basal, cycle 3 and end of induction (EOI). (**A**): M-MDSCs; (**B**): G-MDSCs; (**C**): total MDSCs. * and **, statistically significant differences compared with basal (*p* ≤ 0.05 and *p* ≤ 0.01, respectively). ###, statistically significant differences compared with NVD (*p* ≤ 0.001).

**Figure 4 cancers-13-04622-f004:**
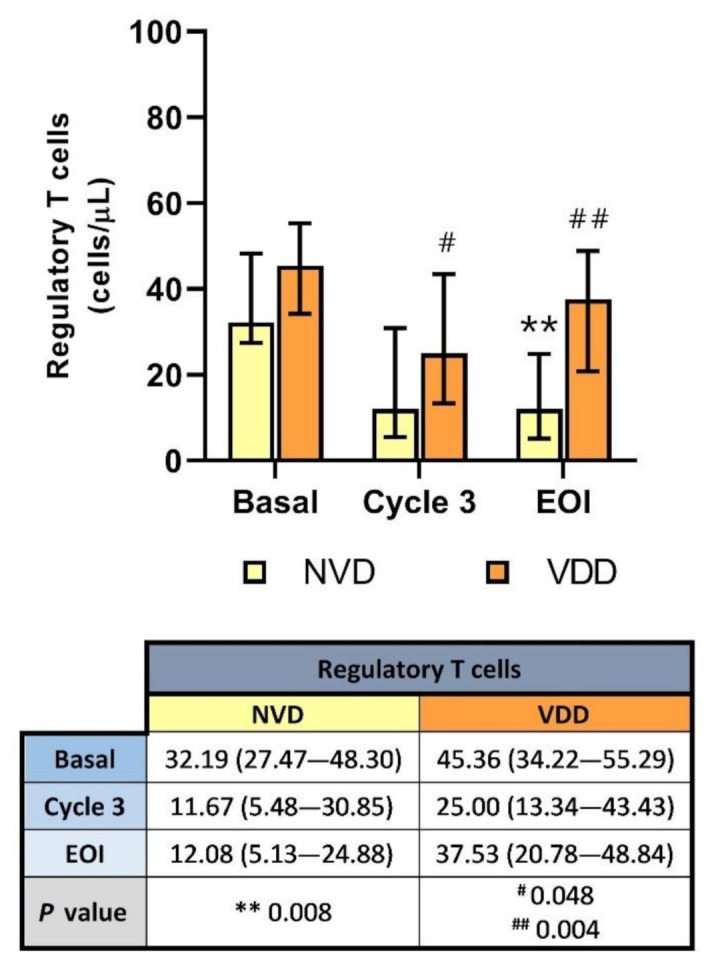
Circulating Treg concentration according to normal vitamin D (NVD) levels and vitamin D deficiency (VDD) in basal, cycle 3 and end of induction (EOI). **, statistically significant differences compared with basal (*p* ≤ 0.01). # and ##, statistically significant differences compared with NVD (*p* ≤ 0.05 and *p* ≤ 0.01, respectively).

**Figure 5 cancers-13-04622-f005:**
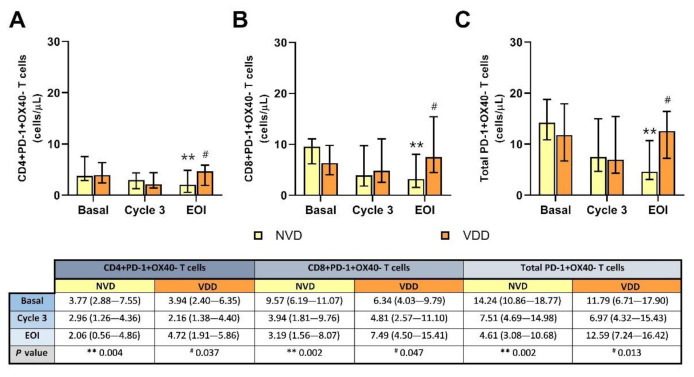
Circulating PD-1+OX40- T cell concentration according to normal vitamin D (NVD) levels and vitamin D deficiency (VDD) in basal, cycle 3 and end of induction (EOI). (**A**): CD4+; (**B**): CD8+; (**C**): total T cells. **, statistically significant differences compared with basal (*p* ≤ 0.01). #, statistically significant differences compared with NVD (*p* ≤ 0.05).

**Figure 6 cancers-13-04622-f006:**
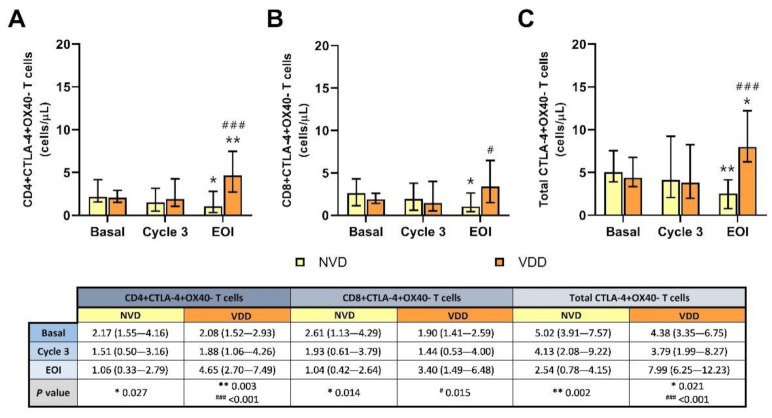
Circulating CTLA-4+OX40- T cell concentration according to normal vitamin D (NVD) levels and vitamin D deficiency (VDD) in basal, cycle 3 and end of induction (EOI). (**A**): CD4+; (**B**): CD8+; (**C**): total T cells. * and **, statistically significant differences compared with basal (*p* ≤ 0.05 and *p* ≤ 0.01, respectively). # and ###, statistically significant differences compared with NVD (*p* ≤ 0.05 and *p* ≤ 0.001, respectively).

## Data Availability

Data are available upon reasonable request.

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
