# Peer review of "Lower Survival and Increased Circulating Suppressor Cells in Patients with Relapsed/Refractory Diffuse Large B-Cell Lymphoma with Deficit of Vitamin D Levels Using R-GDP Plus Lenalidomide (R2-GDP): Results from the R2-GDP-GOTEL Trial"

_cancers, 2021, doi:10.3390/cancers13184622_

Round 1
Reviewer 1 Report
The study is well designed; it addresses the emerging issue of serum ViT D deficiency and DLBCL aggressiveness. The study results support the concept that Vit D deficit may be involved in the immune deficit of patients with relapsed/refractory DLBCL.
A recent study investigated the association of Vit D blood levels and gene expression profile in 2 groups of patients with DLBCL and classic Hodgkin lymphoma (reference Hematol. Oncol. 2021, 39(2): 205-214. We should suggest the Authors to discuss this referenced paper results, in view of the results of the present study.
Author Response
We thank the reviewer for the positive consideration of the manuscript. Following the suggestion of the reviewer, we have included the new reference (reference 65) and further discussed this point.
Reviewer 2 Report
This is a well-conducted and clearly presented study with an impact on the understanding of mechanisms of treatment resistance and potentially on patient management.
Some minimal questions or suggestions:
-Figure 1: I suggest to remove "R2-GDP-COTEL" in the box of HD subjects and it would be useful to mention especially in the legend of this figure the meaning of the abbreviations
-The age of the patients is missing. have you observed a correlation between vitamin D levels and age and excluded, if low levels are seen in older patients, an impact of age itself ? Do you think you can exclude this bias?
Figure 2b represents the level of vitamine D in patients with OS > and < to 24 months. Could you show survival data in VDD versus NDD ?
To be discuss: why the vitamine D level is lower in patients before Ré-GDP versus healthy donors ?
To further investigate these results, it would be necessary to determine whether vitamin D substitution can affect circulating suppressive populations and impact prognosis. To be discussed on already available in vitro data.
Number of references is little too high in my opinion
Author Response
-Figure 1: I suggest to remove "R2-GDP-GOTEL" in the box of HD subjects and it would be useful to mention especially in the legend of this figure the meaning of the abbreviations.
Response: Following the recommendation of the reviewer, we have changed the Figure 1.
-The age of the patients is missing. have you observed a correlation between vitamin D levels and age and excluded, if low levels are seen in older patients, an impact of age itself? Do you think you can exclude this bias?
Response: Age is already included in the manuscript (Material and Methods). We performed a bivariate analysis (Spearman test) and correlation was not obtained between age and vitamin D levels (r=0.091; p=0479)
Figure 2b represents the level of vitamin D in patients with OS > and < to 24 months. Could you show survival data in VDD versus NVD?
Response: Here we provide the survival graph in VDD vs. NVD patients (p=0.540).
To be discuss: why the vitamin D level is lower in patients before R-GDP versus healthy donors?
Response: We have further discussed the differences in Vit D levels and the possible consequences under Discussion section.
To further investigate these results, it would be necessary to determine whether vitamin D substitution can affect circulating suppressive populations and impact prognosis. To be discussed on already available in vitro data.
Response: Following the suggestion of the reviewer, we have further discussed this point with available in vitro data, and we have included a new reference in the Conclusions section (Reference 66).
We are grateful to the reviewer for the help in improving our manuscript
